# Investigation into Micropollutant Removal from Wastewaters by a Membrane Bioreactor

**DOI:** 10.3390/ijerph16081363

**Published:** 2019-04-16

**Authors:** Mohanad Kamaz, S. Ranil Wickramasinghe, Satchithanandam Eswaranandam, Wen Zhang, Steven M. Jones, Michael J. Watts, Xianghong Qian

**Affiliations:** 1Ralph E. Martin Department of Chemical Engineering, University of Arkansas, Fayetteville, AR 72701, USA; makamaz@email.uark.edu; 2Department of Biomedical Engineering, University of Arkansas, Fayetteville, AR 72701, USA; seswara@gmail.com (S.E.); xqian@uark.edu (X.Q.); 3Department of Civil Engineering, University of Arkansas, Fayetteville, AR 72701, USA; wenzhang@uark.edu; 4Garver, 5251 DTC Parkway, Suite 405, Greenwood Village, CO 80111, USA; SMJones@GarverUSA.com; 5Garver, 14160 N Dallas Parkway, Suite 850, Dallas, TX 75254, USA; MJWatts@GarverUSA.com

**Keywords:** activated sludge, biological treatment, endocrine disrupting compounds, trace organic compounds, water reuse

## Abstract

Direct potable reuse of wastewater is attractive as the demand for potable water increases. However, the presence of organic micropollutants in industrial and domestic wastewater is a major health and environmental concern. Conventional wastewater treatment processes are not designed to remove these compounds. Further many of these emerging pollutants are not regulated. Membrane bioreactor based biological wastewater treatment has recently become a preferred method for treating municipal and other industrial wastewaters. Here the removal of five selected micropollutants representing different classes of emerging micropollutants has been investigated using a membrane bioreactor. Acetaminophen, amoxicillin, atrazine, estrone, and triclosan were spiked into wastewaters obtained from a local wastewater treatment facility prior to introduction to the membrane bioreactor containing both anoxic and aerobic tanks. Removal of these compounds by adsorption and biological degradation was determined for both the anoxic and aerobic processes. The removal as a function of operating time was investigated. The results obtained here suggest that removal may be related to the chemical structure of the micropollutants.

## 1. Introduction

The increasing need for direct water reuse requires that wastewater treatment facilities comply with more restrictive effluent regulations, aimed at reducing or eliminating the adverse effects of trace organic compounds on human health [1]. Direct potable reuse refers to the introduction of treated municipal wastewater directly into the municipal water system after appropriate treatment and monitoring. Direct potable reuse is currently used in a limited number of facilities, the most notable examples being Windhoek, Namibia, Big Springs (TX, USA), Cloudcroft (NM, USA) and Fountain Valley (CA, USA).

The presence of organic micropollutants in industrial and municipal wastewater has become a major concern [2]. These emerging pollutants include pharmaceuticals, personal care products, industrial chemicals, pesticides, fire retardants, etc. Discharge guidelines for many of these compounds do not exist. Given that most existing wastewater treatment technologies were not designed for removal of these compounds they could survive in the treated wastewater. If the treated wastewater were directly introduced into the municipal water system, these emerging contaminants will be present in the drinking water. Thus, there is a great need to understand the fate of these compounds and to determine whether adequate clearance is provided in existing wastewater treatment facilities.

This contribution focuses on micropollutants that can affect the human endocrine system. More than 70,000 chemicals are found to have endocrine-disruptive potential [3]. Endocrine disrupting compounds originate from a variety of sources including pesticides, personal care products, antibiotics and pharmaceutically active compounds (PHACs), natural hormones and other man-made chemical compounds [4]. These compounds can interfere with endocrine or hormone systems of humans above a threshold dose. The viability of direct potable reuse at a particular site depends on the effectiveness of the wastewater treatment process in removing endocrine disrupting compounds and other micropollutants that may be present, to levels that are not hazardous for human consumption. Here we investigate clearance using a membrane bioreactor (MBR) treatment system.

Conventional activated sludge (CAS) treatment processes are designed to remove solids, natural organic compounds, and pathogens. The elimination of organic micropollutants using activated sludge is generally not sufficient due to the quantity and variability of these compounds [5]. The levels of removal reported in the literature range from a few percent for recalcitrant compounds to 100% for readily biodegradable molecules or easily adsorbed hydrophobic molecules [6,7,8,9,10]. Moreover, organic micropollutants are often present in trace concentrations making detection and measurement challenging.

Though multiple degradation/removal pathways exist for a variety of compounds depending on their specific molecular structures and properties, the dominant removal mechanisms are adsorption and biodegradation with a minor pathway through volatilization [11]. In particular, electron-donating or electron-withdrawing properties of the functional groups, the acidity or pK_a_ value of the acidic groups and the overall hydrophobicity of the molecule play a major role in the removal mechanisms by the activated sludge process. Hydrophobic compounds (log D > 3.2) demonstrated consistently high removal, over 85%, due largely to adsorption whereas moderately hydrophobic (log D < 3.2) compounds with strong electron withdrawing functional groups showed removal of less than 20% [5,6]. The charge on the compound also affects the level of removal. Depending on the pK_a_ of the acid groups on the micropollutant and the pH of the wastewater, these contaminants can be neutral or charged which will affect both adsorption and biodegradation and hence removal from the wastewater.

Due to the large number and range of molecular structures and properties of the micropollutants present in water, considerable uncertainty remains regarding the level of removal in CAS processes [12]. Further, it appears that a number of xenobiotic compounds are recalcitrant to biodegradation [4]. Since conventional wastewater treatment processes fail to sufficiently eliminate these contaminants, emergent technologies should be considered as alternatives [13]. A membrane bioreactor (MBR) combining the activated sludge process with a membrane barrier for wastewater treatment is one of the technologies that demonstrates several advantages over CAS processes. These advantages include smaller footprint, high effluent quality with complete retention of particulate matter, stable operation with long solid retention times (SRT); high concentrations of mixed liquor suspended solids (MLSS) and low food to microorganism ratios compared to CAS processes [14].

MBR-based treatment steps show potential as a cost-effective method for clearance of many micropollutants from wastewaters [15,16,17,18,19,20]. The MBR is normally operated with a long solids retention time (SRT) leading to higher diversification of microbial culture and potentially enhanced micropollutant removal. Thus, compounds with an intermediate removal efficiency in CAS treatment could generally be further reduced by 20–50% in a MBR [5]. Earlier MBR studies on micropollutant removal focused largely on the aerobic redox condition [19]. A recent study [12] showed that extended anaerobic conditions could enhance biodegradation of micropollutants, some of which were recalcitrant in aerobic MBR operation.

Besides MBR processes, other membranes and membrane processes have been widely used for wastewater treatment. Micropollutants can be removed by direct membrane filtration processes, such as low-pressure microfiltration (MF), ultrafiltration (UF), and high pressure nanofiltration (NF) and reverse osmosis (RO) [21,22,23,24,25]. So far, no single method has demonstrated the capability to remove all the micropollutants completely. Thus, it is the sum of the removal from different unit operations that will determine the clearance for a given wastewater treatment facility. Here we focus on the MBR as there is a significant promise for micropollutant removal from wastewaters. However, no clear removal trends that relate operating parameters and molecular properties to removal for the micropollutants exist.

Several previous studies have quantified removal of micropollutants in wastewater treatment facilities [5,17,20]. Adsorption and biodegradation can be affected by multiple factors and can occur competitively or synergistically. Thus, our focus here is to investigate the clearance of a set of representative micropollutants: acetaminophen, amoxicillin, atrazine, estrone, and triclosan under realistic yet controlled conditions. We combine anoxic and aerobic redox conditions in a laboratory scale MBR with recirculation of aerobically treated water back to the anoxic tank in order to mimic actual operation. Micropollutant removal was conducted after targeted levels for chemical oxygen demand (COD), ammonia-nitrogen (NH_3_-N) and nitrate-nitrogen (NO_3_-N) were reached to establish the proper microbial culture for wastewater treatment. Moreover, the micropollutants selected for this study were subjected to both aerobic and anoxic conditions. The five representative micropollutants were spiked into actual wastewater to give a final concentration of 1 part per million (ppm). We then follow the change in concentration of the five micropollutants over time. Our study is one of a few studies [26] that investigates the degradation of representative micropollutants under recirculating anoxic and aerobic conditions. Moreover, we attempt to quantify the relative amounts of removal by adsorption and biodegradation.

## 2. Materials and Methods

### 2.1. Materials

Amoxicillin trihydrate, triclosan (99%), and estrone (99+%) were purchased from Alfa Aesar (Ward Hill, MA, USA), while acetaminophen (≥98%) and atrazine (≥97%) were procured from Tokyo Chemical Industry (TCI, Chuo-ku, Tokyo, Japan). Liquid phenol, sodium nitroprusside dihydrate (≥98%), sodium hydroxide, sodium hypochlorite and ethanol were purchased from VWR (Radnor, PA, USA) and used as received with no further purification. HPLC grade acetonitrile from EMD Millipore (Bedford, MA, USA) and deionized (DI) water (Milli-Q, 18.2 MΩ cm) were employed as the mobile phase for HPLC analysis. Chemical oxygen demand (COD) kits were purchased from CHEMetrics (Midland, VA, USA) with a range of 0–1500 ppm. Nitrate nitrogen (NO_3_-N) detection kits were purchased from HACH (Loveland, CO, USA).

### 2.2. Selection of Micropollutants

Five organic micropollutants were selected based on the class of compound (antibiotic, herbicide etc.) and abundance in municipal wastewater in the Arkansas–Oklahoma region [25]. The micropollutants tested are given in Table 1. They include pharmaceutically active compounds (acetaminophen, amoxicillin), herbicides (atrazine), steroid hormones (estrone), and personal care products (triclosan). Arkansas and Oklahoma are largely agricultural states with abundant animal and crop farms where it is common to give animals antibiotics and use herbicides on crops. As a result, amoxicillin and atrazine were chosen as two of the micropollutants for this investigation. In addition, the analgesic drug acetaminophen, was chosen as the city of Fayetteville and vicinity has close to half million people. Acetaminophen is one of the most common pharmaceuticals. Estrone was chosen as a representative hormone. Triclosan is widely used as an antimicrobial agent in soaps, detergents, cosmetics and many other personal care products.

Table 1 lists the molecular weight, chemical formula, log D at pH 8 and water solubility of the micropollutants investigated here. Amoxicillin, acetaminophen and atrazine have been detected at 0.03, 0.1, 0.02 ppm levels respectively in the wastewaters used in this study. All compounds were stored at room temperature except for amoxicillin, which was stored at 4 °C. In accordance with their water solubility, acetaminophen and amoxicillin were dissolved in water before spiking whereas atrazine, estrone, and triclosan were dissolved in ethanol/water mixture 1:1 v/v followed by sonication to create a homogenous mixed solution before spiking into the wastewater.

### 2.3. Detection of Micropollutants

High Performance Liquid Chromatography 1260 Infinity HPLC from Agilent Technologies (Santa Clara, CA, USA) equipped with a Luna C_18_ column from Phenomenex (5 μm, size 250 × 4.6 mm, Torrance, CA, USA) was used to measure the micropollutant concentration. The mobile phase was a mixture of acetonitrile and DI water at a flow rate of 0.75 mL/min, with a linear gradient varying from 10 to 100% acetonitrile during the 35 min run followed by 5 min DI water. The column temperature was maintained at 29 °C and the sample injection volume was 20 μL. A diode array detector (DAD) was used for detection. An initial scan ranging from 194 to 270 nm was performed for each compound and the wavelength exhibiting highest sensitivity was selected. Prior to each HPLC run, samples were centrifuged for 5 min at 1000 rpm followed by filtration through a 0.05 μm syringe filter to remove any suspended particulate matter. Detection limits for triclosan and the four other micropollutants were 12.5 ppb and 5 ppb, respectively. Table 2 shows the HPLC detection limits for the five compounds and the corresponding wavelengths for the measurement.

### 2.4. Membrane Bioreactor

A schematic representation of the laboratory MBR system used here is shown in Figure 1. It consisted of anoxic and aerobic tanks (each 35 L) and a microfiltration membrane. The tank in which the microfiltration membrane was placed had a volume of 20 L. The polyvinylidene difluoride (PVDF) microfiltration membrane was provided by Lantian Corporation (Lantian Inc., Yixing, China). It had a nominal pore size of 0.08 μm and an effective surface area of 0.1102 m^2^. The aerobic tank was continuously aerated with a sparger whereas the anoxic tank had a mechanical mixer to provide homogenous mixing (without aeration). In order to better control the laboratory scale MBR, a separate filtration tank was included which had a sparger that supplied coarse bubbles to suppress membrane fouling. Recirculation of the MLSS between the anoxic and aerobic tanks ensured nitrification and denitrification at the two different redox potentials. The aerobic and filtration tanks were not connected directly. However, after reaching the desired operating time, MLSS was transferred from the aerobic tank to the filtration tank using a peristaltic pump.

Wastewater after primary screening was obtained from the West Side Wastewater Treatment Facility (Fayetteville, AR, USA). The treatment plant came on line in 2008. The treatment plant is designed to process an average daily flow of 10 million gallons per day (MGD), a maximum daily flow of 32 MGD. The wastewater obtained for this work contains mainly dissolved organic matter and nutrients (carbon, nitrogen and phosphorus), and is largely free of suspended solids. Activated sludge was collected from both the anoxic and aerobic units of the plant and immediately seeded into our laboratory MBR.

The sludge was loaded into the MBR and was fed with the actual wastewater. It was allowed to acclimatize for up to three weeks until stable nutrient removal as indicated by COD, total ammonia nitrogen (TAN) and NO_3_-N was achieved. Dissolved oxygen (DO) levels in the anoxic and aerobic tanks were carefully monitored to ensure establishment of the two redox zones, which promote nitrification (aerobic) and denitrification (anoxic). The DO level in the anoxic tank ranged from 0.38 to 0.91 ppm whereas it ranged from 2.3 to 3.67 ppm in the aerobic tank depending on the water level and the MLSS circulation rate. The SRT was kept at 35 days and the MBR was operated over a 4-month period.

Samples from the aerobic and anoxic tanks were collected at various times and analyzed. COD, TAN, NO_3_-N, DO and total suspended solids (TSS) were monitored during the experiments. DO was measured using a SympHony^TM^ dissolved oxygen probe VWR International. COD and NO_3_-N were determined according to the protocols given by the manufacturers of the kits. TAN was determined using an established method [27] with ±0.02 accuracy.

Once the targeted levels of COD (<30 ppm), TAN (<5 ppm) and NO_3_-N (<5 ppm) were reached, the five micropollutants at 1 ppm concentration with respect to the total volume (anoxic and aerobic tanks) were added to the anoxic and aerobic tanks following the procedure described below. The 5 micropollutants were first spiked into two 10 L volumes of freshly collected wastewater and well mixed. These two 10 L volumes of spiked wastewater were then fed into the anoxic and aerobic tanks containing 17 L of MLSS respectively to bring the total volume to 27 L for each tank and the concentration of spiked micropollutants to 1 ppm.

The concentration of the micropollutants was determined at various times by withdrawing 50 mL samples. After spiking the 10 L volumes of wastewater with the five micropollutants a sample was withdrawn within 5 min of spiking. After addition of the 10 L volumes of the wastewater containing the micropollutants to the anoxic and aerobic tanks of the MBR, a second sample was withdrawn (from each tank) within 5 min. This is denoted as the time zero sample. Further samples were removed from the anoxic and aerobic tanks after 4, 8 and 12 h of operation. Finally, after 12 h of operation, MLSS from the aerobic tank was transferred to the filtration tank. Filtrate or effluent was then collected and the concentration of the micropollutants determined. In order to maintain the desired SRT (35 days), a fraction of the sludge from both tanks was wasted each week. The recirculation rate of MLSS between the aerobic and anoxic tanks was kept at 32 mL/min using a peristaltic pump. The initial MLSS concentrations were about 4200 and 5400 ppm in the anoxic and aerobic tanks respectively. The MLSS concentrations increased to about 5000 and 6500 ppm respectively at the time of micropollutant addition. Each run was repeated three times under the same operating conditions and the reported results are the average for three runs. For each run two samples were taken for each data point and analyzed with HPLC. Thus, each result represents the average of 6 readings. The error bars provide the range for these six readings.

## 3. Results and Discussion

### 3.1. Overall Performance of MBR

Wastewater quality parameters were monitored daily to evaluate the overall performance of the MBR. The COD in the influent wastewater (from the West Side Wastewater Treatment Facility) ranged from 155–754 ppm and decreased to 10–22 ppm in the MBR effluent (recovered filtrate from the submerged membrane). The TAN decreased from 20–40 ppm in the wastewater from the West Side Wastewater Treatment Facility to less than 1 ppm in the effluent. The NO_3_-N increased up to 20 ppm in the aerobic tank due to nitrification, and eventually decreased to less than 1 ppm in the effluent due to denitrification after recirculating the water between the aerobic and anoxic tanks. COD, TAN and NO_3_-N in the wastewater from the West Side Wastewater Treatment Facility, before addition of the micropollutants (i.e., in the anoxic and aerobic tanks after acclimatization of the sludge) after addition of micropollutants, at the end of the 12-h MBR operation and in the effluent (filtrate) are shown in Table 3. The initial high NO_3_-N concentration in the anoxic and aerobic tanks indicates that nitrification had occurred.

Figure 2, Figure 3 and Figure 4 show the variation of COD, TAN, and NO_3_-N for one of the MBR runs (after addition of the micropollutants). The TSS levels in the anoxic and aerobic tanks reached 5100 and 6500 mg/L respectively. Samples were taken from both tanks after 4, 8 and 12 h (end of run) operation. After 12 h, MLSS from the aerobic tank was transferred to the filtration tank and the effluent collected and analyzed.

The COD of the wastewater before MBR treatment was below 500 ppm. However, its value increased to 1700 and 1500 ppm in the aerobic and anoxic tanks respectively after spiking the micropollutants. This increase was largely due to the addition of ethanol that was used to dissolve the low solubility micropollutants. After 4, 8 and 12 h operation, the COD level in the aerobic tank reduced to around 400, 170 and 24 ppm while the COD level in the anoxic tank reduced to 570, 500 and 350 ppm. The percentage removal of COD in both anoxic and aerobic tanks is shown in Figure 2. After 12-h operation, the COD removal reached over 90% and about 40% in aerobic and anoxic tanks respectively. It is clear that aerobic process is more efficient in reducing COD than the anoxic one. After 12-h operation, the MLSS from the aerobic tank was transferred to the filtration tank. The COD level in the feed to the submerged membrane was 16 ppm and in the effluent was 8 ppm. The membrane was capable of decreasing the COD by about 50%.

The initial TAN in the wastewater was around 25 ppm. After the addition of the micropollutants, TAN levels in the aerobic and anoxic tanks were reduced to 8 and 11 ppm respectively due to the dilution that occurs when 10 L of micropollutant containing wastewater was added. The TAN level in the aerobic tank reduced rapidly to below 1 ppm due to nitrification after 4 h operation and then fell below detection limits. However, due to recirculation of the MLSS between the aerobic and anoxic tanks, TAN levels in the anoxic tank reduced somewhat slowly to about 9.7, 6.3 and 4.6 ppm after 4, 8 and 12 h of operation respectively. The percentage TAN removal in the anoxic and aerobic tanks is shown in Figure 3. It can be seen that nitrification occurs in the aerobic tank with complete TAN removal while 60% TAN removal is obtained after 12 h in the anoxic tank.

Figure 4 shows the denitrification efficiency of NO_3_-N in both tanks. The oxidation or nitrification of TAN to NO_3_^−^ occurs in the aerobic tank while the reduction or denitrification of the NO_3_^−^ to N_2_ takes place in the anoxic tank so the amount of NO_3_^−^ in anoxic tank reduces faster than that of the aerobic tank. After 4 h, about 45% of the nitrate was reduced in the anoxic tank whereas only 15% was reduced in the aerobic tank as shown in Figure 4. The removal of nitrate in the aerobic tank is mainly due to MLSS circulation between the two tanks. NO_3_-N reduced to 5.3, 4.5 and 3.0 ppm after 4, 8 and 12 h of operation with the initial concentration before addition of the micropollutants being 6.3 ppm. The concentration of NO_3_-N in the anoxic tank reduced to 2.0, 1.6 and 1.4 ppm after 4, 8 and 12 h of operation with the initial concentration of 3.6 ppm before addition of the micropollutants. The effluent from membrane filtration showed no change in NO_3_-N indicating little rejection by the membrane. Figure 2, Figure 3 and Figure 4 show that after addition of micropollutants, the reduction in COD, TAN, and NO_3_-N over a 12-h period are as expected indicating proper operation of the MBR.

### 3.2. Micropollutant Removal

The removal of organic matter increases with HRT and SRT in both the aerobic and anoxic tanks which are important parameters that are directly linked to micropollutants removal. For very hydrophobic or rapidly biodegradable compounds, HRT is probably not as important. There is some uncertainty as to the role of SRT on micropollutant removal where longer SRTs lead to the more diversification of the microbial culture thereby potentially improving the degradability of certain recalcitrant compounds. However, the general consensus is that SRT of over 15 days is needed in order to establish a healthy microbial community for micropollutant degradation. The SRT for our current study was kept at 35 days to ensure that the microbial community is sufficiently established and diversified to remove spiked micropollutants. Since our MBR was operated under a semi-continuous mode without a constant wastewater inflow/outflow, the operating time of the micropollutants with the sludge instead of HRT is a more appropriate parameter to describe the removal of these micropollutants.

The wastewater used here, contained very low concentrations of amoxicillin, acetaminophen and atrazine ~100 ppb or less. Nevertheless, all micropollutants were added to the anoxic and aerobic tanks to give a final concentration of 1 ppm (ignoring any low levels occurring in the wastewater). Besides biodegradation, some of the more hydrophobic compounds can be adsorbed by the particulate matter in the wastewater as well as the sludge. The micropollutants can be absorbed by bacterial lipids and the fat fraction of the sludge via hydrophobic interaction [17]. If positively charged, they also bind to the negatively charged polysaccharides outside the bacterial cell membranes. Finally, adsorption can also occur via affinity interactions. As a result, the initial concentration at the beginning of MBR operation varies for different compounds. We assume adsorption is fast relative to biodegradation. In addition, if there is little adsorption the rate of biodegradation is likely to be very slow. Triclosan with a log D value over 4.5 was completely adsorbed by particulate matter in wastewaters and by the sludge. Its concentration at time 0 (within 5 min of addition of the 10 L volumes of wastewater containing the micropollutants to the anoxic and aerobic tanks) was not measurable. Estrone with a log D of over 3.68 was also partially adsorbed. Adsorption of amoxicillin, acetaminophen and atrazine was relatively low in the range of 15–30%.

Figure 5 shows the percentage removal of the micropollutants as a function of operating time. The removal that occurs before time 0, start of bioreactor operation, is due to adsorption onto particulate matter in the wastewater and the sludge. The results are the average for three runs. Two samples were taken for each data point. Thus, each result represents the average of 6 readings. The error bars provide the range for these 6 readings. The TSS for each of the runs was 6040, 6500, 6610 mg/L for the aerobic sludge and 5370, 5090 and 4610 mg/mL for the anoxic sludge respectively.

At time zero, removal of the micropollutants has occurred to varying degrees. The top and bottom panels in Figure 5 show removal through microbial degradation for the anoxic and aerobic tanks. Within the first 4 h of operation over 90% of the amoxicillin and acetaminophen were degraded in the anoxic and aerobic tanks. After 12-h operation both compounds were removed to below detection limits. In the case of estrone, up to 90% was removed by absorption at time 0. However, the remaining 10% was slowly degraded in both the anoxic and aerobic tanks. Total removal was close to 98% after 12 h of operation as shown in Figure 5. However, in the case of atrazine while there was just over 20% removal by adsorption at time zero very little biodegradation was observed over 12-h operation.

Assuming adsorption occurs rapidly compared to biodegradation, the difference between the calculated concentration that was spiked into the 10 L volumes (1 ppm) and the values obtained for samples that were withdrawn immediate upon spiking will provide the amount adsorbed by particulate matter in the wastewater. The difference between the concentration measured at time zero and the calculated concentration that was spiked into the 10 L volume of wastewater will give the amount adsorbed by the biomass and wastewater in the anerobic and aerobic tanks before biodegradation. The results are shown in Figure 6.

Since the wastewater contained organic and inorganic particulate matter, adsorption of the micropollutants occurs as shown in Figure 6. The level of adsorption varied from a few percent to about 50% percent of the total amount spiked. One interesting observation is that the level of adsorption by the particulate matter in the wastewater does not correlate with the hydrophobicity of the compound. Amoxicillin has the lowest hydrophobicity, but it shows the highest adsorption by the particulate matter in the wastewater. Triclosan has the highest hydrophobicity, but displays very low adsorption. Acetaminophen was also poorly adsorbed. About 30% of the atrazine and estrogen are adsorbed by the particulate matter in the wastewater. The adsorption of these micropollutants by the particulate matter in the wastewater does not follow entropy-driven hydrophobic interaction. It seems that enthalpic changes play a more important role. Amoxicillin with many functional groups for potential hydrogen bonding interaction demonstrated the highest adsorption. In addition, amoxicillin is acidic and possesses a negative charge at neutral pH. Electrostatic interaction also likely plays an important role in adsorption by the particulate matter in the wastewater.

When the 10 L volume of wastewater containing micropollutants was fed into the aerobic and anoxic tanks, additional adsorption of the compounds by the sludge occurred rather rapidly. Figure 6 also shows the removal by adsorption at time zero in the anoxic (top) and aerobic (bottom) tanks respectively. Triclosan was almost completely adsorbed by the sludge in both tanks. It is the most hydrophobic of the micropollutants selected, with log D value over 4.5. This is followed by estrone the second most hydrophobic compound. Over 90% of the estrone was adsorbed. Little additional adsorption of acetaminophen was observed. In the case of amoxicillin and atrazine the overall removal by adsorption decreased indicating release of the adsorbed micropollutants. This confirms that the adsorption of these compounds in wastewater only is not due to hydrophobic interaction, rather via affinity or charge mediated interaction process.

Figure 7 shows that percentage removal by biodegradation based on the concentration present at time zero. Biodegradation is rapid for amoxicillin and acetaminophen in both aerobic and anoxic tanks. Over 95% was removed in 4 h of operation. However, biodegradation of atrazine is minimal. The level of biodegradation of atrazine in this study is in agreement with previously reported values in in literature [28]. Atrazine was found to be non-biodegradable by both anoxic and aerobic microbial culture.

The biodegradation of estrone shows an interesting phenomenon. Biodegradation of estrone is rather slow. Even after a 12 h of operation, estrone remained in both the aerobic and anoxic tanks at a concentration of about 20 ppb or higher. It also appears that aerobic sludge is more efficient in digesting estrone. Hu et al. [29] studied the removal of hormones and their conjugates using pilot-scale and lab-scale MBRs operated with raw wastewater. They found that the removal of estrone ranged between 80.2–91.4% in pilot and lab scale MBR systems [29].

The different percentage removal of the five micropollutants can be correlated with their molecular structures and properties. Very hydrophobic compounds can be easily removed due to adsorption by the sludge as demonstrated here for triclosan and estrone. Hydrophilic compounds possessing electron-donating groups such as amoxicillin and acetaminophen are readily biodegradable. Compounds with intermediate hydrophobicity and containing electron withdrawing groups are generally not prone to biodegradation as shown here for atrazine. In the case of estrone, it is largely removed via hydrophobic adsorption to the sludge, it is also biodegraded as it does possess electro-donating groups. Overall the molecular structure of the micropollutants will have a strong influence on their removal in a MBR.

Table 4 compares removal of the five micropollutants in literature studies with the removal obtained here. As can be seen removal of the micropollutants has been investigated in both real and synthetic wastewaters. In the case of acetaminophen, amoxicillin, estrone, and triclosan, our results are in excellent agreement with previous studies that used real wastewater irrespective of the scale of the MBR. Further most earlier studies considered aerobic operation only. Removal of these compounds from synthetic wastewaters is lower. Synthetic wastewaters are likely to contain a lower diversity of microorganisms as well as lower levels of suspended solids resulting in the lower observed removal. The results indicate that the laboratory scale MBR system we have developed models actual MBR systems accurately.

Table 4 indicates that atrazine removal is much lower than the other compounds in agreement with our results. The fact that removal in actual wastewater (studied here) is in between removal levels observed for synthetic wastewaters obtained in literature studies suggests that biodegradation of atrazine is limited. This is in agreement with our results. While the results obtained here clearly indicate the feasibility of micropollutant removal by an MBR process containing both anoxic and aerobic tanks, further studies should be conducted using a large MBR that operates in truly continuous mode thus better modeling actual industrial practice.

## 4. Conclusions

Experiments were performed to investigate the removal of five selected micropollutants from wastewater using a recirculating MBR system consisting of an anoxic tank, an aerobic tank and separate membrane filtration unit. It was found that the MBR is efficient and effective at removing four of the five compounds: amoxicillin, acetaminophen, estrone, and triclosan. Atrazine is recalcitrant with only about 5% removal by biodegradation in both redox conditions and about 25% adsorption by the particulate matter in the sludge. It was found that operation time is critical for the removal of estrone as estrone biodegrades slowly. Triclosan was found to be rapidly adsorbed by both anoxic and aerobic sludge. Amoxicillin and acetaminophen demonstrated relatively low adsorption but rapid biodegradation. The adsorption by the particulate matter in the wastewater appears to be affinity or electrostatically mediated whereas adsorption by the sludge is largely dictated by the hydrophobicity of the compounds.

## Figures and Tables

**Figure 1 ijerph-16-01363-f001:**
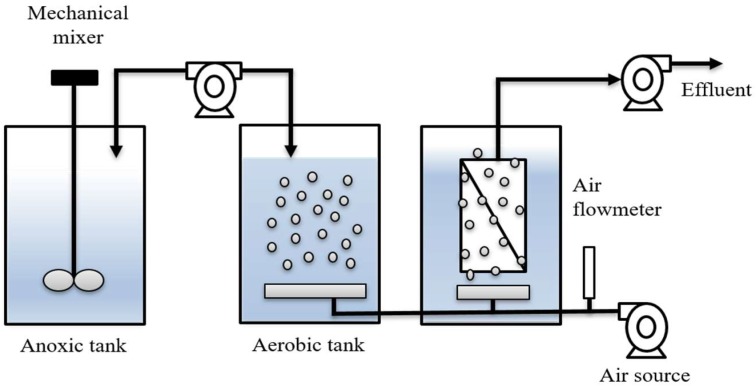
Schematic diagram of membrane bioreactor consisting of an anoxic tank, an aerobic tank and a separate membrane filtration tank. Recirculation of the MLSS (mixed liquor suspended solids) between the tanks ensured nitrification and denitrification at the two different redox potentials.

**Figure 2 ijerph-16-01363-f002:**
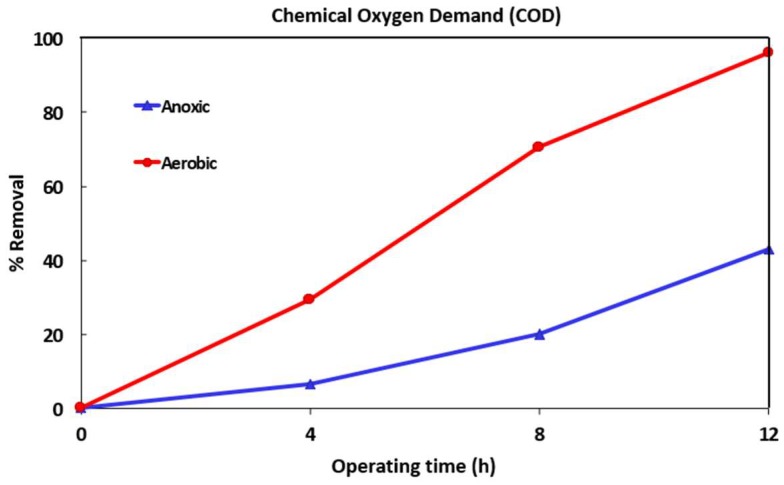
Percentage COD removal during 12-h MBR operation. TSS (total suspended solids) in the anoxic and aerobic tanks was about 5100 and 6500 mg/L respectively.

**Figure 3 ijerph-16-01363-f003:**
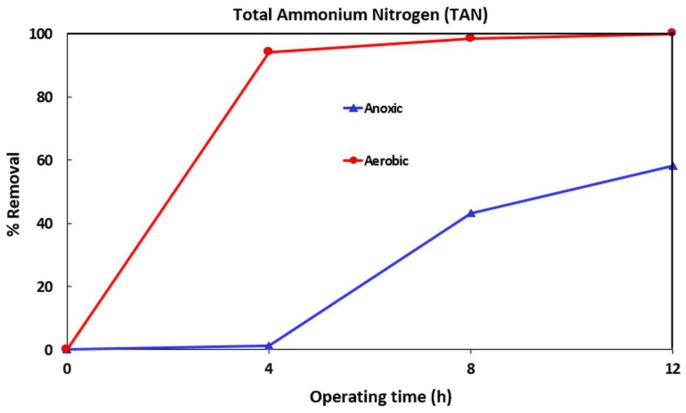
Parentage TAN removal during 12 h MBR operation. TSS in the anoxic and aerobic tanks was about 5100 and 6500 mg/L respectively.

**Figure 4 ijerph-16-01363-f004:**
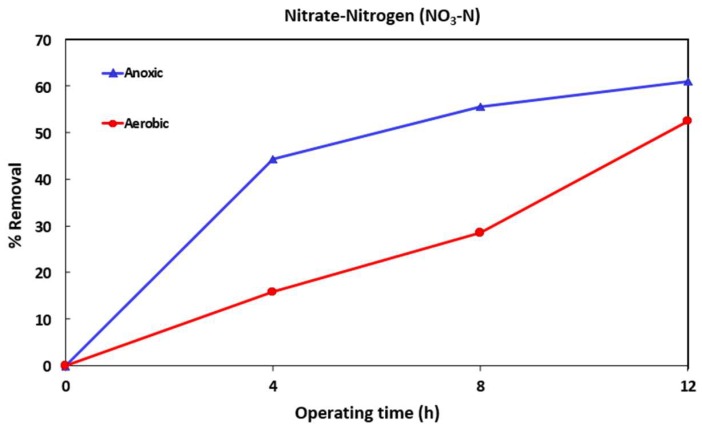
Percentage NO_3_-N removal during 12 h MBR operation. TSS in the anoxic and aerobic tanks was about 5100 and 6500 mg/L respectively.

**Figure 5 ijerph-16-01363-f005:**
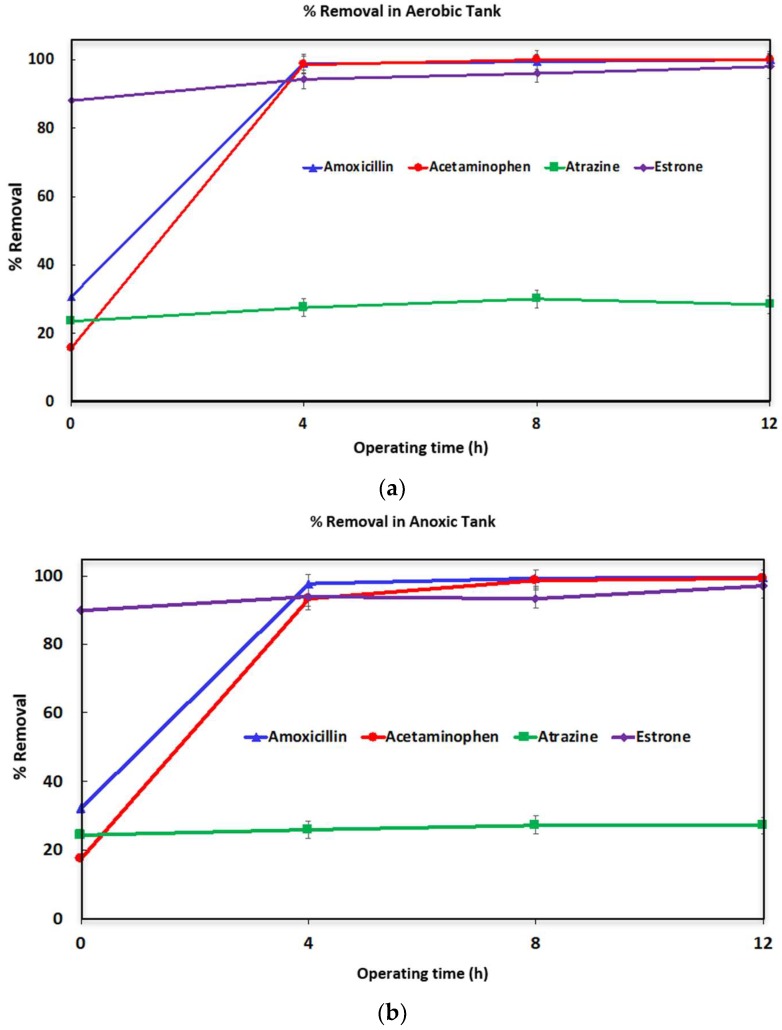
Percentage removal of the micropollutants in the aerobic (**a**) and anoxic (**b**) tanks as a function of operating time. Experiments were conducted at room temperature, 25 °C.

**Figure 6 ijerph-16-01363-f006:**
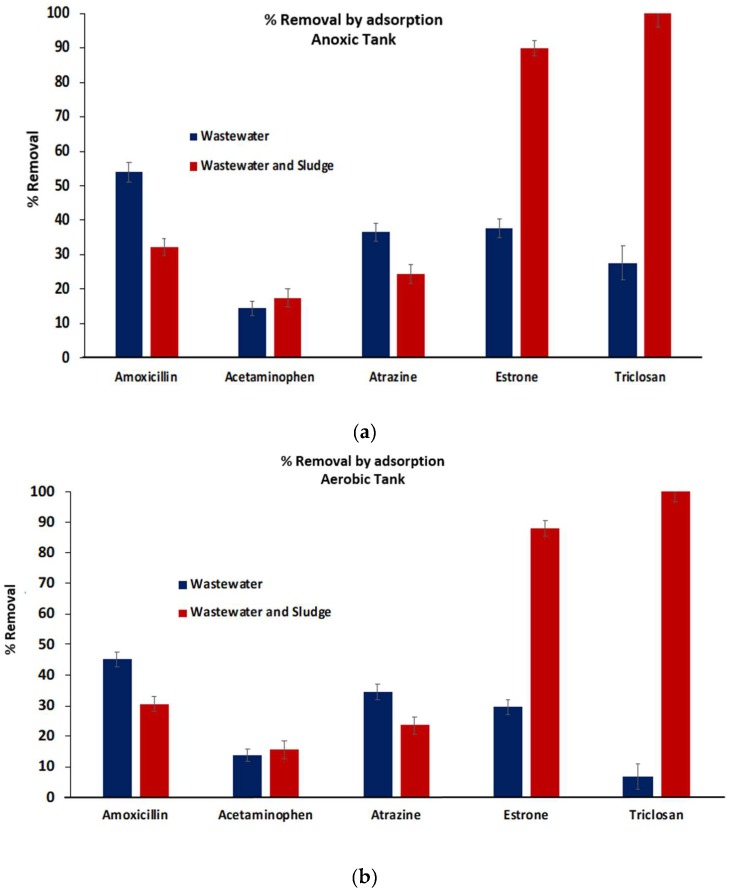
Percentage removal of the micropollutants by adsorption in the 10 L volume of wastewater (blue) and wastewater with sludge (red) in anoxic (**a**) and aerobic (**b**) tanks respectively.

**Figure 7 ijerph-16-01363-f007:**
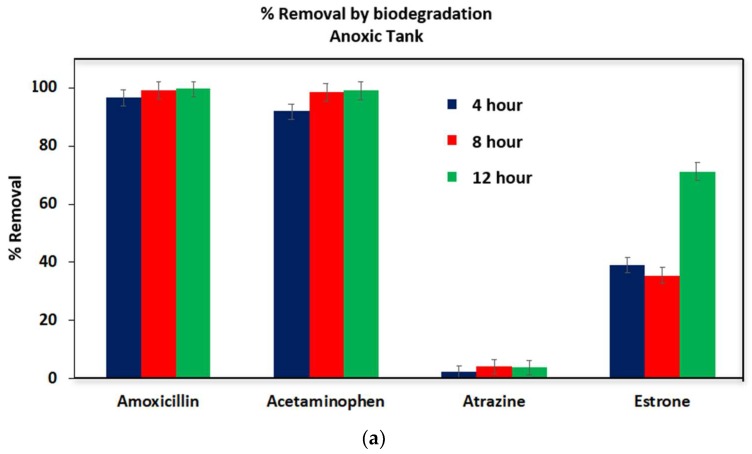
Percentage removal by biodegradation of the micropollutants in anoxic (**a**) and aerobic (**b**) tanks. The reference concentrations for the compounds were those measured at time zero.

**Table 1 ijerph-16-01363-t001:** The structure and physicochemical properties of the micropollutants.

Compound	Molecular Weight (g/mol)	Structure	Log D (pH = 8)	Water Solubility (ppm)
Amoxicillin, Antibiotic (C_16_H_19_N_3_O_5_S)	365	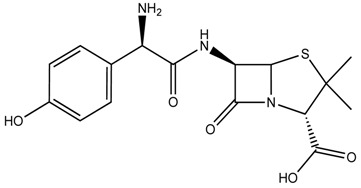	−2.56	3.4 × 10^3^
Acetaminophen, Pharmaceutical (C_8_H_9_NO_2_)	151	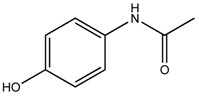	0.33	1.4 × 10^4^
Atrazine, Herbicide (C_8_H_14_ClN_5_)	216	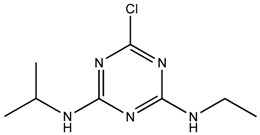	2.63	34.7
Estrone, Hormone (C_18_H_22_O_2_)	270	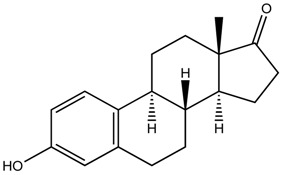	3.68	30
Triclosan, Antibacterial C_12_H_7_Cl_3_O_2_	290	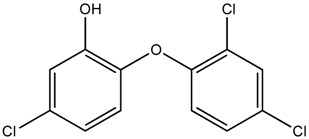	4.76	10

**Table 2 ijerph-16-01363-t002:** The detection limits of five selected compounds.

Compound	Detection Limit (ppb)	Detection Wavelength (nm)
Amoxicillin	5	198
Acetaminophen	5	198
Atrazine	5	222
Estrone	5	194
Triclosan	12.5	198

**Table 3 ijerph-16-01363-t003:** Wastewater quality parameters for the initial wastewater received from the treatment plant, the anoxic and aerobic tanks before addition of the micropollutants and at the beginning and end of 12-h MBR (membrane bioreactor) run, and the effluent.

Wastewater Quality Parameter	COD (ppm)	TAN (ppm)	NO_3_-N (ppm)
Initial wastewater from treatment plant	484	25.6	12.7
Anoxic tank before micropollutant addition	578	8.4	3.6
Aerobic tank before micropollutant addition	614	11.1	6.3
Anoxic tank after 12 h operation	350	4.6	3.0
Aerobic tank after 12 h operation	24	0	1.4
Effluent	8	0	1.6

**Table 4 ijerph-16-01363-t004:** Comparison of micropollutant removal obtained here with literature studies.

Micropollutants	Process	Removal (%)	Reference
Acetaminophen	Pilot-MBR, actual wastewater	100	[30]
Lab-scale MBR, synthetic wastewater	95	[31]
This study	100	
Amoxicillin	Pilot-MBR, synthetic wastewater	77	[32]
MBR, actual wastewater	100	[33]
This study	100	
Atrazine	Lab-scale MBR, synthetic wastewater	40	[34]
Lab-scale MBR, synthetic wastewater	8	[26]
This study	<25	
Estrone	Lab-scale MBR, synthetic wastewater	>90	[18]
Pilot-MBR, synthetic wastewater	88	[32]
Pilot-MBR, actual wastewater	95–100	[35]
This study	98	
Triclosan	Lab-scale MBR, synthetic wastewater	>90	[36]
MBR, actual wastewater	98	[37]
This study	100

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
