# Peer review of "Investigation into Micropollutant Removal from Wastewaters by a Membrane Bioreactor"

_ijerph, 2019, doi:10.3390/ijerph16081363_

Round 1
Reviewer 1 Report
Manuscript No.: ijerph-474295
Title: Investigation into Micropollutant Removal from Wastewaters by a Membrane Bioreactor
Reviewer’s comments
First, the reviewer would thank the editor for inviting me. The study is about micropollutant removal from wastewater by MBR. Through investigating the removal of representative micropollutants in an semi-continuous MBR through adsorption and biodegradation, the authors found that the removal could be related to the chemical structure and properties. The experiments were generally well designed and suitable for the journal. The findings are interesting to the readers of the journal. However, there are some minor issues to take care of before publishing. Therefore, the reviewer suggests major revision.
Major:
1. The introduction presents the background and necessity of the study well. however, it is a bit lengthy. The authors may need to make it more succinct. In addition, the novelty of the study should be justified.
2. Methodology. Perhaps, I missed it. How did the authors control the quality of data? Did they repeated the experiments or analyses? What replication? Were the aerobic tank and the membrane tank connected? How?
3. The authors assumed that within 5 min, the removal of micropollutants was caused by physical adsorption. It could be true. It would be more convincing if a separate batch experiment is performed to comparing biodegradation and adsorption. For instance, examining the physical adsorption of micropollutants onto flocs by inactivating the microorganisms using certain biocidal agent such as NaN3.
4. What are the limitations of this study? What would the authors suggest being done in the future?
Minor:
1. Please pay attention to inconsistent tense. Some mistakes throughout the context.
2. Please include citations for some statements. Such as P9L21, the sentence “The micropollutants can be …”
3. Hours vs. h. Sometimes the authors use h, sometimes hours. Please be consistent.
4. P13L11, intermediate hydrophobic with electron withdrawing. Maybe change with to and?
5. P1L37, punctuation. Semi-colon could be used after country names.
6. P2L44, more diversification? Or higher diversification?
7. P3L7-8, consider revising the sentence.
8. P6L37, add dash between 12 and h.
9. P6L30, add a space between numbers and dash of the range notation.
Reviewer 2 Report
The work is clear and well written and organized. I think that the work can be improved if some additional results is reported about the performance of the membrane (flux, fouling etc) since the system is an MBR. I suggest also some additional comparison with the existing litarature related to MBR
Reviewer 3 Report
This is an important research on the possible removal of chosen micropollutants from wastewater, by using membrane bioreactor. The manuscript is worth to be published however the improvement of the content must be performed. The list of abbreviation must be added to the text. On page 6, lines 14-17 - the described procedure is unclear. It looks that Authors added micropollutants twice (?). The quality of figures 2-5 should be improved. Authors should also refer to the results presented by other authors. Final recommendation - minor revision.
Round 2
Reviewer 1 Report
The reviewer is glad to see the modification made by the authors. The presentation and quality of the manuscript are now much better. Recommend a YES.